# Nutrient Deficiency Promotes the Entry of *Helicobacter pylori* Cells into *Candida* Yeast Cells

**DOI:** 10.3390/biology10050426

**Published:** 2021-05-12

**Authors:** Kimberly Sánchez-Alonzo, Fabiola Silva-Mieres, Luciano Arellano-Arriagada, Cristian Parra-Sepúlveda, Humberto Bernasconi, Carlos T. Smith, Víctor L. Campos, Apolinaria García-Cancino

**Affiliations:** 1Laboratory of Bacterial Pathogenicity, Department of Microbiology, Faculty of Biological Sciences, Universidad de Concepcion, Concepción 4070386, Chile; kimsanchez@udec.cl (K.S.-A.); fabiolalsilva@udec.cl (F.S.-M.); lucarellano@udec.cl (L.A.-A.); cparras@udec.cl (C.P.-S.); csmith@udec.cl (C.T.S.); 2Laboratorio Pasteur, Concepción 4030000, Chile; hbernasconi@lpasteur.cl; 3Laboratory of Environmental Microbiology, Department of Microbiology, Faculty of Biological Sciences, Universidad de Concepción, Concepción 4070386, Chile; vcampos@udec.cl

**Keywords:** *Helicobacter pylori*, nutrient deficiency, starvation, intracellular *H. pylori*, *Candida*, endosymbiosis, fetal bovine serum

## Abstract

**Simple Summary:**

*Helicobacter pylori* is a pathogenic bacterium which causes several gastric and extra-gastric pathologies in humans. This pathogen is capable of entering eukaryotic cells of humans or of other species, including yeasts of the genus *Candida*. These yeasts are resistant to stressing environmental conditions (such as pH changes or scarce nutrients) which threaten the viability of *H. pylori*. Therefore, *Candida* yeasts may harbor this bacterium when subjected to stressing conditions or become transmission vehicles for it. Further research is required to establish the stressing environmental factors triggering the harboring of *H. pylori* within yeasts. The present work evaluated if deficiency or absence of the necessary nutrients favors the endosymbiotic relationship between these two microorganisms, facilitating the viability of the bacterium. In fact, a deficiency of nutrients increased the harboring of viable *H. pylori* cells within the yeast cells. On the contrary, in the complete absence of nutrients, the presence of intra-yeast bacteria was reduced. Therefore, yeast cells may contribute to the subsistence of this pathogenic bacterium when subjected to nutrient deficiency until it may infect an appropriate host, such as humans. The present work may also contribute, with further studies, to elucidate the transmission routes used by the pathogen *H. pylori* to infect its hosts.

**Abstract:**

*Helicobacter pylori*, a Gram-negative bacterium, has as a natural niche the human gastric epithelium. This pathogen has been reported to enter into *Candida* yeast cells; however, factors triggering this endosymbiotic relationship remain unknown. The aim of this work was to evaluate in vitro if variations in nutrient concentration in the cultured medium trigger the internalization of *H. pylori* within *Candida* cells. We used *H. pylori*–*Candida* co-cultures in Brucella broth supplemented with 1%, 5% or 20% fetal bovine serum or in saline solution. Intra-yeast bacteria-like bodies (BLBs) were observed using optical microscopy, while intra-yeast BLBs were identified as *H. pylori* using FISH and PCR techniques. Intra-yeast *H. pylori* (BLBs) viability was confirmed using the LIVE/DEAD BacLight Bacterial Viability kit. Intra-yeast *H. pylori* was present in all combinations of bacteria–yeast strains co-cultured. However, the percentages of yeast cells harboring bacteria (Y-BLBs) varied according to nutrient concentrations and also were strain-dependent. In conclusion, reduced nutrients stresses *H. pylori*, promoting its entry into *Candida* cells. The starvation of both *H. pylori* and *Candida* strains reduced the percentages of Y-BLBs, suggesting that starving yeast cells may be less capable of harboring stressed *H. pylori* cells. Moreover, the endosymbiotic relationship between *H. pylori* and *Candida* is dependent on the strains co-cultured.

## 1. Introduction

*Helicobacter pylori* is a bacterial pathogen which infects the gastric epithelium of more than 50% of the world population. The infection caused by this bacterium may produce gastric and extra-gastric pathologies [1,2]. Gastric pathologies include an acute/chronic gastritis present in 100% of the infected population which, when not treated, may progress to more severe conditions, including atrophic gastritis, intestinal metaplasia and, even more severe, gastric cancer [3,4,5]. Since 1994, *H. pylori* has been considered to be a type I carcinogen for humans by the Agency for Research on Cancer [6]. Gastric cancer is the fifth most common cancer in humans and third in cancer-caused deaths worldwide [4]. Although not all *H. pylori* infected persons will develop this type of cancer, the literature reports the close relationship between the infection with *H. pylori* and the progression to gastric cancer [4]. Therefore, the Maastricht V Consensus established that all persons positive for *H. pylori* require antibiotic treatment [7]. Among the extra-gastric pathologies caused by *H. pylori*, it is possible to mention, among others, idiopathic thrombocytopenic purpura and iron deficiency anemia [8,9].

The conditions provided by the gastric environment where *H. pylori* survives are challenging to replicate in the laboratory, making this pathogen a fastidious microorganism to culture in vitro [10,11,12]. Enriched culture media, such as Brucella agar or broth, Columbia agar (CA), brain heart infusion agar, or trypticase soy agar, provide the nutritional requirements for bacterial isolation. Due to its role as an enzymatic cofactor and its role in electron transport, iron is an essential element for the metabolism of organisms [13,14]. Therefore, all culture media to culture *H. pylori* must be supplemented with an iron source such as 5% to 10% horse blood or serum. Fetal bovine serum (FBS) can be used as an alternative (at the same concentrations), and Brucella broth can be supplemented with either 5% to 10% FBS or equine serum [15]. In both cases, supplementation with blood or serum provides the same function, serving as an iron source for *H. pylori* [12,15,16].

Despite the nutritional and environmental demands required by *H. pylori*, this bacterium can be detected using molecular techniques in the oral cavity, amplifying specific genes of this microorganism in dental plaque, saliva, dental pulp, and in 30% of cases of severe cavities in children between 4 and 7 years [17,18,19]. Furthermore, the genes of *H. pylori* have been amplified by PCR from environmental sources such as tap water, insects (e.g., flies and blowflies), raw sheep meat, feces and raw milk from cows, buffaloes and sheep [20,21,22,23]. The diversity of environmental sources from where *H. pylori* genes have been amplified and the difficulty of isolating this bacterium from those sources raise the following questions: how does *H. pylori* DNA reach extra gastric locations? Is the survival of this bacterium possible in extra-gastric niches?

It has been reported that *H. pylori* in extra-gastric environments is capable of varying its morphology from helical to coccoid (viable non-culturable stage) [24,25]. It has also been observed that under specific environmental stimuli such as cold stress, pH, aerobiosis, and aging, its metabolism of ATP, polyphosphate and RNA can vary, and its urease activity decreases [18,19]. Under FBS starvation, the bacterium, in addition to decreasing its metabolism, is capable of forming biofilms to ensure its survival [25,26]. However, if the stress conditions are not reversed in time, bacterial death will occur, and therefore, the question of how *H. pylori* survives in extra-gastric niches remains unanswered. In this sense, further investigation could unravel the intracellular association between *H. pylori* and yeast cells.

Most of the *Candida* species hosted by humans colonize the oral cavity, lungs, skin, stomach, intestine, vagina and urinary tract [27], and they can be vertically transmitted from the mother to the child during birth [28,29]. Yeast cells present in those niches may interact with different bacterial species, including the harboring of bacteria within yeast cells [27], and facilitate the bacterial dissemination throughout the host.

Yeast cells harboring bacteria-like bodies (Y-BLBs) have been isolated from different niches, such as the oral cavity of adults and newborns, gastric biopsies, vaginal discharge from pregnant women, fruits, flowers, different foods, and insects [30,31,32,33,34,35,36]. These observations led to the suggestion that the vacuoles of fungal cells constitute a specialized survival niche for *H. pylori*, providing protection to this pathogen when it is subjected to environmental stressing conditions [37]. This proposal is further supported by assays co-culturing *H. pylori* and *Candida albicans* under low pH conditions (pH 3–4), which demonstrated a higher percentage of yeast cells harboring this bacterium in co-cultures at acidic pH when compared to controls at neutral pH [38]. This result is indicative of a higher entry of *H. pylori* into *C. albicans* under pH conditions unfavorable for the bacteria [38]. Based on these findings, it is necessary to evaluate other environmental factors, such as nutrition conditions, that may promote the internalization of *H. pylori* into yeast cells and influence bacterial transmission and protection. Therefore, the aim of the present work was to evaluate in vitro if variations in FBS concentration in the culture medium can trigger the internalization of *H. pylori* into *Candida* cells.

## 2. Materials and Methods

### 2.1. Bacterial Strains and Growth Conditions

Four *Candida* strains were used in this study. These strains were the reference strains *C. albicans* ATCC 90028 and *Candida glabrata* ATCC 90030 and the clinical strains *C. glabrata* LEO-37 (obtained from the oral cavity) and *C. albicans* VT-3 (obtained from vaginal discharge). Four *H. pylori* strains were used: three of them were reference strains SS1, G-27 and J99 (also known as ATCC 700824), and the fourth one was the clinical strain H707 (obtained from a gastric biopsy). Yeast cells were cultured in Sabouraud agar (SA) (Merck, Darmstadt, Germany) plus chloramphenicol (CHL) (OXOID, Basingstoke, UK) following the manufacturer’s instructions. Yeast cultures were incubated at 37 °C for 24 h in an incubator under aerobiosis (ZHICHENG, Shanghai, China). Bacterial strains were grown in Columbia agar (CA) (OXOID, Basingstoke, UK) supplemented with 5% FBS (Biological Industries, Cromwell, CT, USA) (CA-5%FBS) and incubated under microaerobic conditions (10% CO_2_ and 5% O_2_) at 37 °C for 48 h to 72 h (Thermo Scientific, Waltham, MA, USA). 

To confirm the purity of cultures, verification tests were performed. In the case of *Candida*, Gram-staining and urease tests were performed and colonies were randomly seeded on CRHROM agar. On the other hand, for *H. pylori*, Gram-staining and urease, catalase and oxidase tests were made. 

### 2.2. Growth Curves at Different Concentrations of FBS for H. pylori and Candida Strains 

The growth curves were obtained using Brucella broth (BB) (Difco, Wokingham, UK) supplemented with different concentrations of FBS (1%, 5%, or 20%) or in the absence of nutrients using 0.89% saline solution (SS). To start the incubation to determine the growth curves, *H. pylori* and *Candida* strains were adjusted to an optical density (O.D.) of 0.1 at 600 nm at time zero. Then, 200 µL of each strain suspension was placed in 96-well plates (Thomas Scientific, Swedesboro, NJ, USA) and incubated under microaerobic conditions in Infinite M200 PRO equipment (TECAN, Männedorf, Switzerland). The microaerobic condition was obtained using CampyGen sachets (Thermo Scientific, Waltham, MA, USA). The absorbance of *H. pylori* cultures was measured every 8 h for 72 h, while the absorbance of yeast cultures was measured every 2 h for 50 h. Absorbances were measured at 600 nm using the same Infinite M200 PRO equipment. All assays were performed in triplicate.

### 2.3. Co-Cultures of H. pylori Strains with Candida Strains

Individual suspensions of *Candida* and *H. pylori* strains were prepared in SS at an O.D. of 0.1 at a wavelength of 600 nm. Four mL of BB supplemented with 1%, 5% or 20% FBS or 4 mL of SS was placed in wells of 12-well plates (Thomas Scientific, Swedesboro, NJ, USA), and then 500 µL of each strain was added to each well to obtain the co-cultures. The co-cultures were incubated at 37 °C for 48 h under microaerobic conditions in an incubator (Thermo Scientific, Waltham, MA, USA). Each co-culture was repeated thrice.

### 2.4. Search for Bacteria-Like Bodies (BLBs) and Culture of Yeast Cells Containing BLBs

Fresh co-cultures were prepared as described above and 20 µL aliquots were collected from each well at time 0 and then again at 1, 3, 6, 12, 24, and 48 h. Each aliquot was placed on a glass slide and observed using the 100X objective lens of an optical microscope fitted with a camera (Leica, Wetzlar, Germany). The percentage of Y-BLBs was determined after counting 200 yeast cells. 

From each co-culture containing Y-BLBs, a 20 µL aliquot was streaked on Sabouraud agar supplemented with chloramphenicol (SA-CHL), according to the manufacturer’s instructions, and incubated at 37 °C for 24 h in aerobiosis. The purpose of CHL was to eliminate extracellular *H. pylori* cells. At the end of the incubation period, the presence of Y-BLBs was verified observing, by optical microscopy, wet mounts from randomly selected colonies obtained on SA-CHL and the absence of extracellular bacterial contamination was confirmed using Gram-staining. Colonies were randomly selected and added to 1 mL of BB-5%FBS, mixed using a vortex (DLAB, Ontario, CA, USA), 0.015 µL/mL clarithromycin was added and the culture was incubated under microaerobic conditions at 37 °C for 24 h to eliminate any possible remaining extracellular *H. pylori* cells. After incubation, 20 µL of the culture was taken and spread on SA-CHL by surface dissemination and subsequently incubated at 37 °C for 24 h under aerobic conditions. Colonies were reseeded 6 times in SA-CHL in order to make certain that no extra yeast bacterial cells remained in the culture medium.

### 2.5. Identification of BLBs within Yeast Cells Using Fluorescent In Situ Hybridization (FISH) Technique

Pure cultures of *H. pylori* J99 and *C. albicans* ATCC 90028 strains were used as controls. The presence of Y-BLBs was verified by means of wet mounts of yeast cells obtained from SA-CHL colonies. For the FISH technique, we modified the protocol of Böckelmann and coworkers [39]. Colonies were taken at random from *Candida* cultures where Y-BLBs were observed and resuspended in sterile 1X phosphate buffered saline (PBS) (Sigma-Aldrich, St. Louis, MO, USA) until reaching a turbidity similar to tube 3 of the McFarland scale. Tubes with yeast cell suspensions were centrifuged at 6700× *g* for 2 min (Eppendorf, San Diego, CA, USA) and the supernatant discarded; this step was repeated once more. After discarding the supernatant, a second time, 1X PBS was added to the pellet and vortexed until the entire pellet was resuspended. One hundred µL of each suspension was smeared on a slide and allowed to dry for approximately 20 min. Once the samples were dry, they were fixed with 200 µL of 37% formaldehyde solution (Sigma-Aldrich, St. Louis, MO, USA) and incubated at 4 °C in a humid chamber for 3 h. After the formaldehyde was removed and the samples were allowed to dry at room temperature, they were incubated for 3 min each with 50%, 80%, and 96% ethanol concentrations at room temperature. After the final incubation with ethanol, samples were allowed to dry at room temperature for approximately 15 min. Once dry, 100 µL of the hybridization solution (270 µL of 5M NaCl; 30 µL of 1 M TRIS-HCl; 525 µL of 37.7% deionized formamide; 675 µL of MiliQ water; and 1.5 µL of 10% SDS) was added to each one of the smears followed by 6 µL of the Hpy probe 5′-CACACCTGACTGACTATCCCG-3′ labeled with Cy 3 [40] at a concentration of 5 ng/µL. The solution was carefully mixed and incubated in a wet chamber at 46 °C to 48 °C for 90 min in a thermoregulated bath (Elma, Singen, Germany) in the darkness. After hybridization, each smear was washed twice with the washing buffer (700 µL of 5M NaCl, 1 mL of TRIS-HCl, 48.25 mL of sterile distilled water and 50 µL of 10% SDS). One mL of buffer was used per wash and it was left to incubate in a humid chamber in a thermoregulated bath at 46 °C to 48 °C for 20 min the first time and 5 min the second time, discarding the remaining buffer each time. Then, it was allowed to dry and 200 µL of 1:10 aniline blue in 1X PBS (both Sigma-Aldrich, St. Louis, MO, USA) was added and incubated for 10 min. Subsequently, it was washed 2 times with 1 mL of 1X PBS each time, allowing it to dry at room temperature in the dark. Finally, the samples were observed in a fluorescence microscope (Motic, Viking Way, Richmond, BC, Canada) fitted with the FITC (AT480/535) and TRIC (AT540/605) filters. The images were processed using ImageJ software version 1.53 (NIH Image, Bethesda, MD, USA) to combine the images obtained with the Merged filters.

### 2.6. Detection of H. pylori 16S rRNA

This assay was performed using 24 h old cultures in SA medium where the presence of Y-BLBs was observed. Yeast cells were resuspended in 1 mL of SS at an O.D. of 0.1 at 600 nm and centrifuged at 6700× *g* for 2 min (Eppendorf, San Diego, CA, USA). The pellet was resuspended in 1 mL of 10 mM Tris EDTA buffer (TE) pH 8.0, vortexed for 5 s (DLAB, Ontario, CA, USA), centrifuged at 11,300× *g* for 5 min (Eppendorf, San Diego, CA, USA), and the supernatant discarded. In 200 µL of TE buffer, a suspension at an O.D. of 0.1 at 600 nm was obtained and a heat shock was performed by freezing at −80 °C for 30 min followed by thawing in a thermoblock at 100 °C for 10 min. Three cycles were carried out under these conditions. After the last cycle, the tubes were incubated at −80 °C for 24 h. The next day, the samples were incubated at 70 °C for 2 h and the total DNA extracted using the commercial NucleoSpin Tissue kit (MACHEREY-NAGEL, Düren, Germany), as per manufacturer’s instructions. After total DNA extraction, amplification of the *H. pylori 16S rRNA* gene was performed according to Sánchez-Alonzo et al. [38]. Twelve and a half mL of the master mix, 1 µL of primers F-5′CTCGAGAGACTAAGCCCTCC3′ R-5′ATTACTGACGCTGATGTGC3, 5.5 µL of PCR grade water, and 1.5 μL of DNA sample. Thirty amplification cycles were programmed in a thermocycler (Eppendorf, San Diego, CA, USA), which included cycles for initial denaturation temperature at 94 °C/1 min, denaturation temperature at 98 °C/30 s, hybridization temperature at 53 °C/5 s, extension temperature at 72 °C/40 s and a final extension temperature at 72 °C/10 min. Electrophoresis was performed in 2% agarose gel (Lonza, Walkersville, MD, USA) plus 1 µL of GelRed (Biotium, Landing Parkway, Fremont, CA, USA). Five µL of amplified product and 1 µL of molecular weight markers were added to each well of the gel in a separate well. The gel was run at 70 V for 90 min. Subsequently, the amplicons were visualized by exposing the gel to ultraviolet light in the ENDURO model photo documenter (Labnet, Edison, NJ, USA).

### 2.7. Cell Viability Assay

This assay was performed to confirm the viability of *H. pylori* inside the yeast vacuoles. Aliquots of bacteria suspensions containing Y-BLBs were stained with a solution of LIVE/DEAD BacLight Bacterial Viability kit L-7012 (ThermoFisher, Waltham, MA, USA) and incubated for 15 min in darkness. After incubation, the tubes were vortexed (DLAB, Ontario, CA, USA) at its minimal speed for 3 s. Ten µL of the suspension were added to each slide and a coverslip added to visualize each sample under the 100X objective lens of a fluorescence microscope fitted with a camera (Motic, Viking Way, Richmond, BC, Canada). The FITC (AT480/535) and TRIC filters (AT540/605) were used. Images were obtained, processed using ImageJ software (NIH Image, Bethesda, MD, USA) and superimposed to create merged images.

### 2.8. Statistical Analysis

Data were processed using the SPSS 24.0 software (IBM Company, Armonk, NY, USA). Tukey’s test was applied to determine the presence of statistically significant differences. Values of *p* ≤ 0.05 were considered significant and values *p* ≤ 0.0001 were considered highly significant. According to the results of Tukey’s test, data shown in tables or figures sharing the same letter are not significantly different. 

## 3. Results

### 3.1. Strain Cultures

Cultures of *H. pylori* strains were grown in CA-5%FBS until the typical morphology of colonies was obtained, showing small, translucent, colorless, convex-shaped bacterial colonies (Figure 1A). Gram-negative bacilli were observed after a Gram-stain (Figure 1B). Enzymatic activity tests detected the enzymes urease, catalase, and oxidase (Figure 1C–E). Figure 2 depicts the culture and confirmation of yeast purity on SA-CHL, where the growth of whitish, creamy, convex colonies was observed (Figure 2A). The Gram-stain (Figure 2B) showed the typical yeast-like morphology of *Candida* and the absence of extracellular bacteria that could be contaminating the fungal culture. Figure 2C shows a wet mount of yeast cells whose vacuoles lack BLBs (arrows). Lastly, Figure 2D shows a negative result for the urease test, confirming the absence of *H. pylori*. 

### 3.2. Growth Curves of the Different Strains of H. pylori and Candida at Varying FBS Concentrations or SS 

The growth obtained in *H. pylori* strains was directly affected by a low percentage of FBS or its replacement by SS. No significant growth differences were observed in the presence of 5% or 20% FBS, reaching the exponential growth phase from approximately 16 h to 56 h. No growth was observed in the presence of 1% FBS or SS-only (Figure 3).

*Candida* grew in BB supplemented with 1%, 5% or 20% FBS and SS. A more pronounced growth curve was observed in the culture supplemented with 20% FBS. No significant differences were observed when comparing the growth curves in the presence of 1% or 5% FBS. Yeast strains quickly reached a stationary phase in the presence of SS-only cultures (Figure 4).

### 3.3. Search for BLBs within Yeasts of the Genus Candida

In wet mounts of *H. pylori*–*Candida* co-cultures observed using the 100X objective lens of an optical microscope, the presence of BLBs within the vacuoles of yeast (Figure 5A, black arrows) was detected. The movement of BLBs within the vacuoles is shown in Video S1. In addition, we observed an accumulation of coccoid and bacillary bacteria on the yeast pseudohyphae previously co-cultured with *H. pylori* cells (Figure 5A, red arrow). Cells from a pure *Candida* culture showed, as expected, the absence of BLBs (Figure 5B).

When counting the Y-BLBs in the different co-cultures in BB supplemented with 1%, 5% or 20% FBS, the highest percentage of Y-BLBs (53% to 60%) was obtained in the co-cultures where 1% FBS was used, followed by co-cultures supplemented with 5% and 20% FBS (22% to 26% and 7% to 12%, respectively) (Figure 6). On the other hand, the lowest Y-BLBs percentages were observed in the co-cultures incubated in SS-only (Figure 6). 

Subsequently, in order to analyze the interaction of the different strains of *Candida* and *H. pylori* when co-cultured, the time at which the highest Y-BLBs mean was obtained in each co-culture supplemented with the different FBS concentrations was determined (Figure 7). Once the times with the highest percentages of mean Y-BLBs were obtained, the data were analyzed using Tukey’s test to determine if any differences were dependent on the type of bacterial or fungal strains or only on the concentration of FBS. The analysis showed that the most important factor generating the intracellular interaction between *H. pylori* and *Candida* was the concentration of FBS in the medium. However, in all the co-culture conditions, higher means were observed between *H. pylori* strains H707 and J99 with *C. glabrata* ATCC90030 and LEO-37 strains (Figure 8). The interaction of all co-cultures is shown in Appendix A. 

### 3.4. Identification of BLBs in Yeast Using the Fluorescent In Situ Hybridization (FISH) Technique

To identify the mobile BLBs harbored within the vacuoles of yeast cells, the FISH technique, combined with a specific probe, labeled with Cy3 fluorochrome, for *H. pylori* was used. This assay showed the expected red fluorescence confirming that BLBs harbored within yeast cells were, in fact, *H. pylori* (Figure 9A). When this technique was performed in yeast cells obtained from a pure culture, there was no hybridization of the probe specific for *H. pylori* (Figure 9B), but the hybridization took place when testing cells from a pure *H. pylori* culture (Figure 9C).

### 3.5. Detection of H. pylori 16S rRNA Gene

Figure 10 shows the amplification of the *16S rRNA* gene of *H. pylori* in total yeast DNA extracted after incubating a *H. pylori*–*C. glabrata* strains co-culture during 48 h. The expected amplicon, with a size of 110 bp, was observed in the lanes corresponding to DNA extracted from yeast cells previously co-incubated with the bacterium. The absence of amplification in the lane of the negative control (DNA of pure *C. glabrata* ATCC 90030) and the blank (water grade PCR, master mix and primers) confirmed that there was no *H. pylori* contamination of the yeast strains nor of the reagents used.

### 3.6. Cell Viability Assay

After confirming, using FISH and PCR, that BLBs harboring within yeast cells belong to the *H. pylori* species, the viability of intra-yeast bacteria was tested using the LIVE/DEAD BacLigh Bacterial Viability kit. The green fluorescence observed within the vacuole of yeast cells indicates that *H. pylori* cells retained their viability once inside the yeast cells (Figure 11). In addition, the mobility of the bacteria within the yeast vacuole can be inferred by their change in position in time-lapse photography at 1 s intervals (Figure 11).

## 4. Discussion

*H. pylori* is a pathogen adapted to colonize the stomach, an anatomical site rich in nutrients, including minerals such as iron [41]. These nutritional conditions are not so easy to replicate in the culture media used in the laboratory; hence, this bacterium is considered fastidious to culture. Since it contains hemoglobin, cholesterol, proteins and other factors required by *H. pylori* to grow, FBS supplementation of the culture media used for the isolation of this microorganism is an excellent option as a source of nutrients that favors the in vitro growth of this bacterium [12,15].

In this work, it was observed that the growth of *H. pylori* in BB-20%FBS was not significantly different when compared to the bacterial growth obtained using 5% FBS. One of the most abundant components in FBS is iron, and although it is an important growth factor for many organisms, including *H. pylori*, it can also be toxic at high concentrations [13,41]. During digestion, the human stomach, where *H. pylori* naturally resides, is subjected to changes in iron concentration [42].

This pathogen can be found in the gastric mucus, adhered to the gastric epithelium or located intracellularly in the epithelial cells [43,44], locations where iron availability is scarce. When there are high iron concentrations in its environment, *H. pylori* can control the uptake mechanisms of this chemical element, making use of the iron-sensitive protein family known as ferric uptake regulator (FUR). This protein family (which includes FecA1, FecA2, FrpB1, and FeoB) regulates the bacteria’s iron uptake systems, acting as iron transporters. When iron intracellular levels are low, FUR cannot bind to the Fur box’s sequences, considerably increasing the transcription of genes associated with extracellular iron uptake. This increases the exchange of extra and intracellular iron from the host and vice versa [16].

The lack of growth when *H. pylori* was cultured either in BB-1%FBS or in SS was expected because this bacterium typically requires 5% to 10% FBS [12,15,45]. Consequently, the lack of a nutritious culture medium and the absence of FBS as seen in SS cultures were unfavorable growth conditions for *H. pylori*. These findings are supported by prior reports indicating that bacteria under nutritional stress generate a characteristic response that involves the rapid accumulation of guanosine tetraphosphate (ppGpp) and the inhibition of stable RNA synthesis; therefore, bacteria enter a stationary growth phase. This is also true for *H. pylori* under starvation conditions [45,46]. The growth curves reported in this study are similar to those reported in the literature where, under starvation conditions, the SpoT protein restricts bacterial growth but cells maintain their helical shape [45,46]. 

The *Candida* strains analyzed in the present work grew in all FBS concentrations tested, and overgrowth was observed using BB-20%FBS. The growth observed in SS was significantly lower when compared to the curves obtained when *Candida* strains were incubated in FBS-supplemented culture medium. It has been described that yeasts have an excellent capacity to adapt to changes in the environment, and under starvation conditions *Candida* is capable of forming biofilms to survive, but not to replicate [47,48]. The results obtained in this study, when *Candida* strains were cultured in SS, are similar to those of Delgado and coworkers [49]. On the other hand, like *H. pylori*, yeasts have specialized machinery to regulate iron uptake when this element is in high or low extracellular concentration [50].

The first Y-BLBs were identified during the first hour of co-culturing both microorganisms. The maximum percentage of Y-BLBs was observed at 24 h and 48 h in all co-cultures, but the mean of Y-BLBs obtained in the co-culture containing BB-1%FBS was significantly higher than that obtained in the other culture conditions. When analyzing the percentages of Y-BLBs, the figures were more than 50% in BB-1%FBS, between 22% and 26% in BB-5%FBS, 7% to 12% in BB-20%FBS, and 9% in co-cultures carried out in SS. Therefore, the percentages of Y-BLBs increased as the concentration of FBS decreased. This may be due to the stressing conditions sensed by *H. pylori* that threaten its viability. On the contrary, these culture conditions offered *Candida* all the nutrients necessary for its growth, providing protection for *H. pylori* cells capable of harboring within yeast cells. Interestingly, we identified extracellular bacteria that adhered to the pseudohyphae of yeasts, which may be due to the adhesins present on the surface of this fungal structure.

Regarding this last observation, the adherence of *H. pylori* cells to several species of the *Candida* genus was first reported by Ansorg and Schmid [51]. The adherence of several bacteria to the surface of fungi has been reported and, in the particular case of *Candida*, it has been reported for *Staphylococcus aureus*, Group B *Streptococcus*, *Staphylococcus epidermidis* and *Streptococcus gordonii*, among others, perhaps promoting the dissemination of bacteria throughout the host [52,53,54,55]. Regarding the internalization of *H. pylori* in a human cell model, it has been reported that it occurs most often in ulcer-related samples [56]. It is worth noting that the *H. pylori* strain showing the best capacity to harbor within *Candida* was the J99 strain, whose genotype is related to the development of duodenal ulcers. Another work reported the presence of a link between the coexistence of *H. pylori* and *Candida* and gastric ulcers, suggesting the synergism of these microorganisms in the pathogenicity of this disease [57]. 

Considering that both BB-1%FBS and SS are poor in nutrients, it was possible to anticipate that co-cultures in SS might show a percentage of Y-BLBs similar to that of BB-1%FBS, but this was not the case. In fact, the percentage of Y-BLBs in SS was not significantly different to that observed in the co-culture in BB-20%FBS. The absence of nutrients in SS might negatively affect the mobility of *H. pylori* and favor the development of its coccoid stage (immobile) [58]. Although yeast cells were also in the absence of nutrients, they can survive under inanition and modulate their highly dynamic cell wall according to the conditions to which they are subjected [59]. In yeasts such as *C. glabrata* and *C. albicans*, starvation promotes the migration of cytolytic proteins, and they can alter the expression of cell wall proteins. This leads to cell impermeability, adhesion, and the formation of biofilms to survive under stress conditions [49]. In addition, during starvation and other stress conditions that affect *C. albicans*, the chlamydosporulation process is triggered, promoting the generation of chlamydospores, which are cells with a much thicker cell wall, which would make the entry of *H. pylori* difficult [60]. In addition, the loss of mobility and of bacterial viability can explain the low percentage of Y-BLBs found in SS co-cultures. On the contrary, in the abundance of FBS, both microorganisms were not under stress and, therefore, *H. pylori* was not induced to seek refuge within yeast cells, but a baseline entry occurred anyways. 

The identification of BLBs as *H. pylori* was performed using both FISH and PCR techniques. Furthermore, the LIVE/DEAD BacLight Bacterial Viability kit allowed us to confirm the viability of *H. pylori* within fungal cells. These results are similar to our previous works subjecting *H. pylori* cells to other stressing conditions [30,35] and reports by Siavoshi et al. [30,61].

Lastly, we investigated if the endosymbiotic interaction between different *H. pylori* and *Candida* strains was strain dependent. Although there was no significant difference between the means of Y-BLBs obtained by the different strains in the same co-culture conditions, the higher means observed were between *C. glabrata* (ATCC 90030 and LEO-37) strains and *H. pylori* J99 and H707 strains. At the present time, there are still no reports that allow us to comment on the behavior between strains. However, it has been reported that under starvation, *C. glabrata* expresses adhesion cell wall proteins and also that this species possesses a higher percentage of mannoproteins when compared to *C. albicans*. The above could be the basis for future research to answer this question.

## 5. Conclusions

The results of *H. pylori* and *Candida* co-cultures demonstrated that nutrient deficiency (using FBS as a supplement of nutrients model) is a stressor for *H. pylori*, which significantly increased its entry into *Candida* cells. Furthermore, under starvation of both *H. pylori* and *Candida* strains, the percentages of Y-BLBs decreased significantly, suggesting that starving yeast cells may be less capable of harboring stressed *H. pylori* cells. The results obtained in this study also highlight that the endosymbiotic relationship between *H. pylori* and *Candida* is dependent on the bacterial strains. Therefore, based on the results obtained in this study, it can be inferred that yeast cells may contribute to the subsistence of this pathogenic bacterium when subjected to nutrient deficiency until it may infect an appropriate host, such as humans.

## Figures and Tables

**Figure 1 biology-10-00426-f001:**
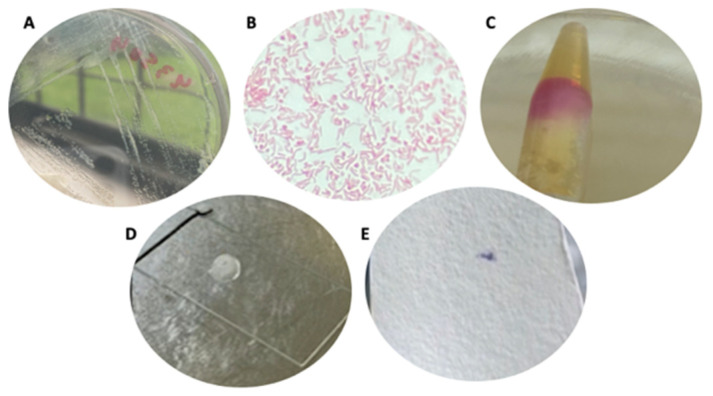
Typical culture and microbiological identification of *H. pylori*. (**A**) Culture of *H. pylori* in CA-5%FBS; (**B**) Gram-stained smear; (**C**) Urease test; (**D**) Catalase test; and (**E**) Oxidase test. CA-5%FBS: Columbia agar supplemented with 5% FBS.

**Figure 2 biology-10-00426-f002:**
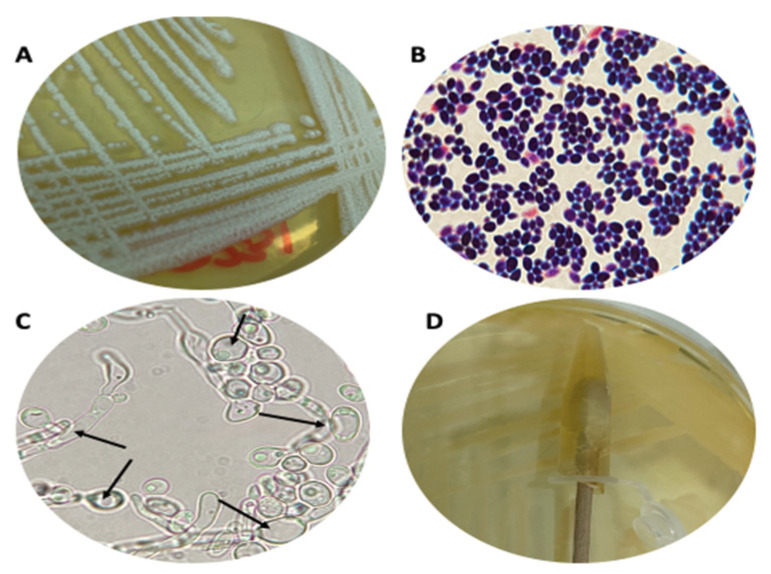
Culture and purity verification of *Candida* ATCC 90030 strain as representative of the strains used in the present work. (**A**) Yeast culture on Sabouraud agar; (**B**) Gram-stain of yeast cells; (**C**) Wet mount of yeast cells obtained from pure cultures. Note the vacuoles of yeast cells (black arrows) lacking BLBs; (**D**) Urease test performed on yeast cultures. BLBs: bacteria-like bodies.

**Figure 3 biology-10-00426-f003:**
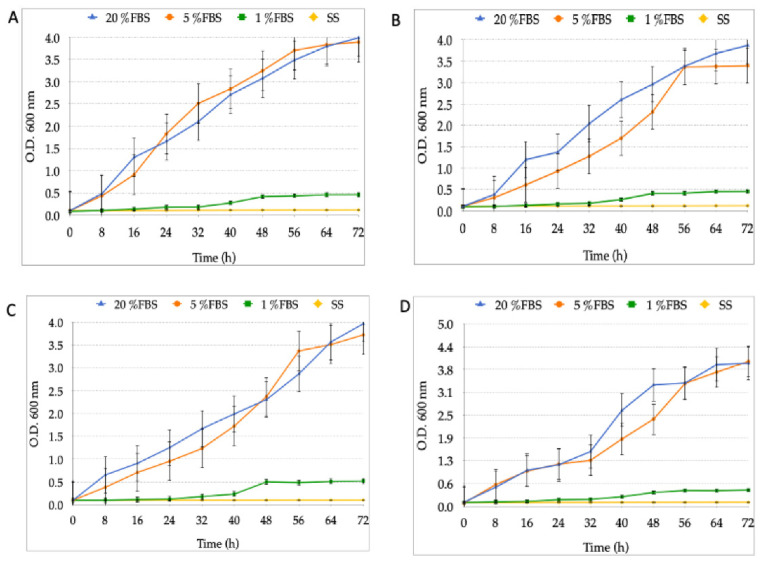
Growth curves of different *H. pylori* strains in the presence of BB supplemented with 1%, 5% or 20% FBS or SS. (**A**) *H. pylori* J99; (**B**) *H. pylori* G-27; (**C**) *H. pylori* SS1 and (**D**) *H. pylori* H707. BB: Brucella broth; FBS: fetal bovine serum; SS: saline solution.

**Figure 4 biology-10-00426-f004:**
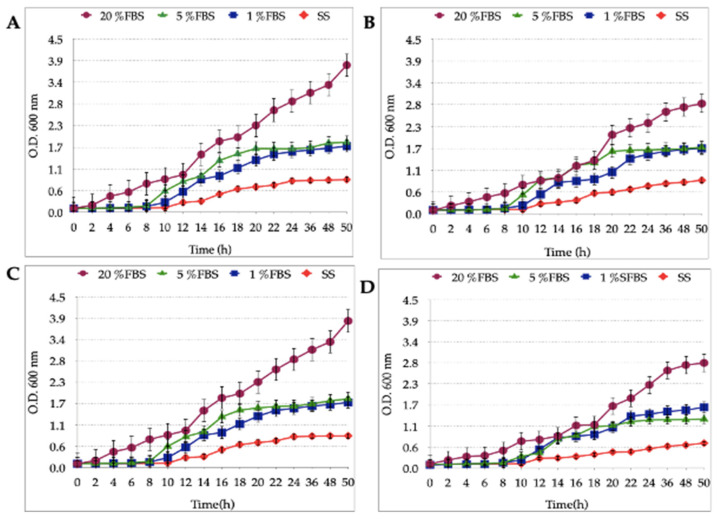
*Candida* growth curves in the presence of BB supplemented with 1%, 5% or 20% FBS or its absence. (**A**) *C. albicans* ATCC 90028; (**B**) *C. glabrata* ATCC 90030; (**C**) *C. albicans* VT-3; (**D**) *C. glabrata* LEO-37. The highest growth level was achieved using 20% FBS, followed by the cultures with 1% and 5% FBS. In the latter, there was no significant difference identified between these two growth curves *p* > 0.05; however, a highly significant difference was found between the growth curves in SS when compared with the other treatments *p* < 0.0001.

**Figure 5 biology-10-00426-f005:**
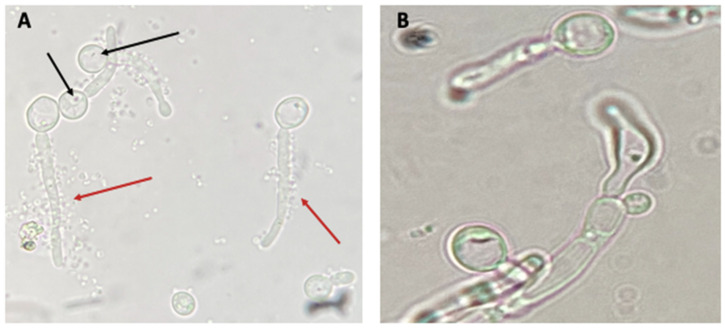
(**A**) Wet mount of *H. pylori* H707–*C. albicans* VT-3 24 h co-culture in BB-1%FBS showing Y-BLBs (black arrows) and *H. pylori* cells attached to pseudohyphae (red arrows). (**B**) Wet mount of *C. albicans* VT-3 cells from a pure culture (negative control) showing the absence of Y-BLBs. Movement of BLBs within yeasts is shown in Video S1. BB-1%FBS: Brucella broth supplemented with 1% of fetal bovine serum.

**Figure 6 biology-10-00426-f006:**
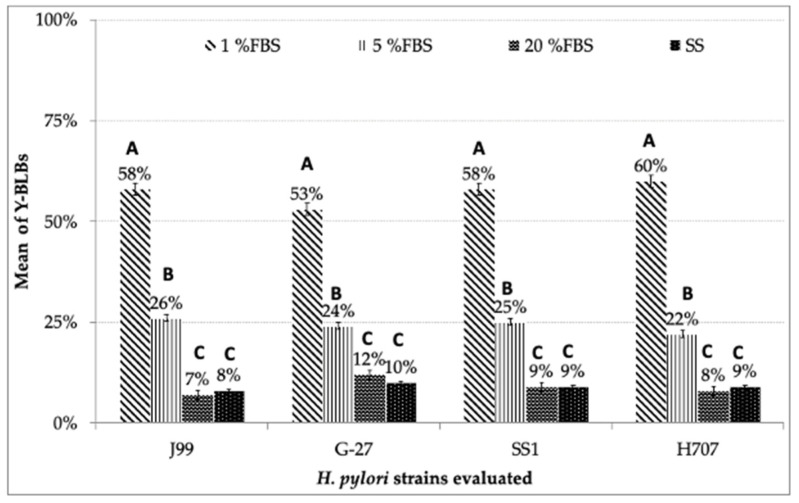
Mean percentage of Y-BLBs when *H. pylori* and *Candida* spp. were co-cultured in BB supplemented with 1%, 5% or 20% FBS concentrations or in SS. Mean percentages of Y-BLBs remained above 50% in the co-cultures supplemented with 1% FBS. No significant differences were observed between the percentages of Y-BLBs obtained in the co-cultures evaluating the same strains of *H. pylori* in the presence of different FBS concentrations. Different letters indicate significant differences (*p* < 0.05).

**Figure 7 biology-10-00426-f007:**
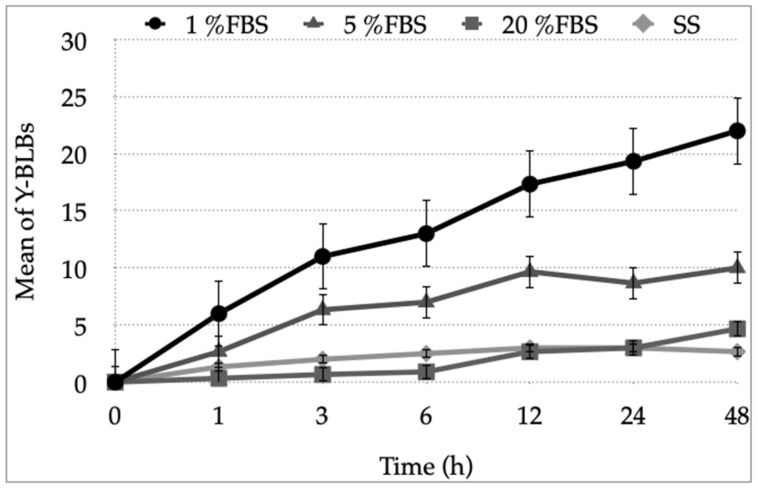
Means of Y-BLBs were obtained for all bacterial strain and yeast strain combinations cultured in BB supplemented with 1%, 5% or 20% FBS or SS during 48 h. This figure shows the means of Y-BLBs obtained when co-culturing *H. pylori* J99 and *C. glabrata* ATCC 90030 strains. The higher means of Y-BLBs for this bacteria–yeast combination were observed at 24 h and 48 h. Culture medium supplemented with 1% or 5% FBS produced the higher means of yeast cells harboring bacteria.

**Figure 8 biology-10-00426-f008:**
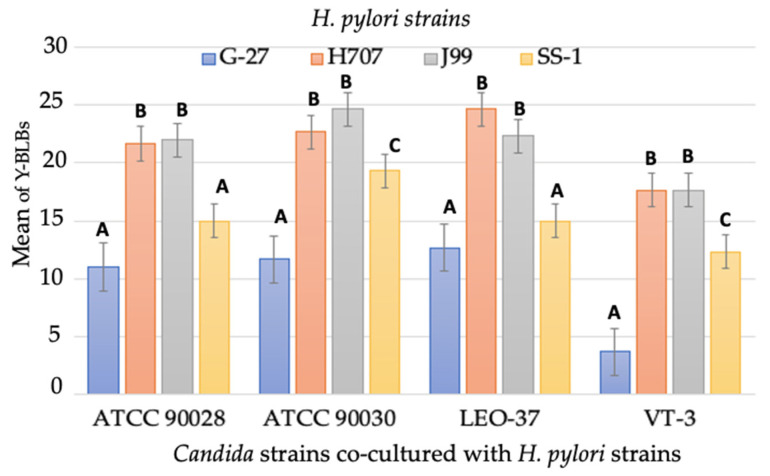
Mean of Y-BLBs identified in the co-cultures incubated in BB-1%FBS for 48 h. There is no significant difference in Y-BLB means when comparing co-cultures carried out with *H. pylori* J99 or H707 strains with all yeast strains. The higher means of Y-BLBs were obtained in the co-cultures carried out with these two strains and the two *C. glabrata* strains. Results are expressed as mean ± SD. Means with different letters are significantly different (*p* < 0.05).

**Figure 9 biology-10-00426-f009:**
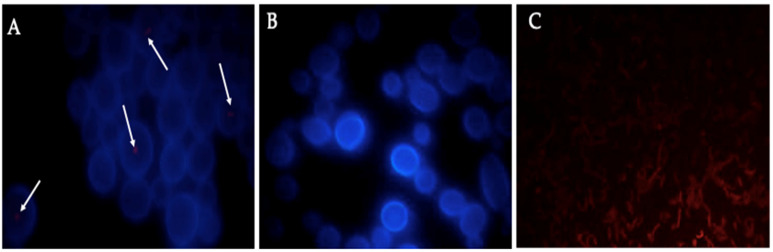
Images showing the results of FISH analysis using a fluorescent probe specific for *H. pylori*. Yeast cells obtained from a *H. pylori* G-27–*C. glabrata* ATCC 90030 strain co-cultured in BB-5%FBS. (**A**) Hybridization of the *H. pylori* specific probe within *C. glabrata* ATCC 90030 cells (white arrows); (**B**) *C. glabrata* ATCC 90030 from a pure culture (negative control) showing the absence of hybridization of the fluorescent probe; (**C**) Pure *H. pylori* G27 strain used as positive control (red fluorescence). Blue fluorescence corresponds to the binding of aniline blue to 1–3ℬ-glucans of yeast cells. BB-5%FBS: Brucella broth supplemented with 5% FBS.

**Figure 10 biology-10-00426-f010:**
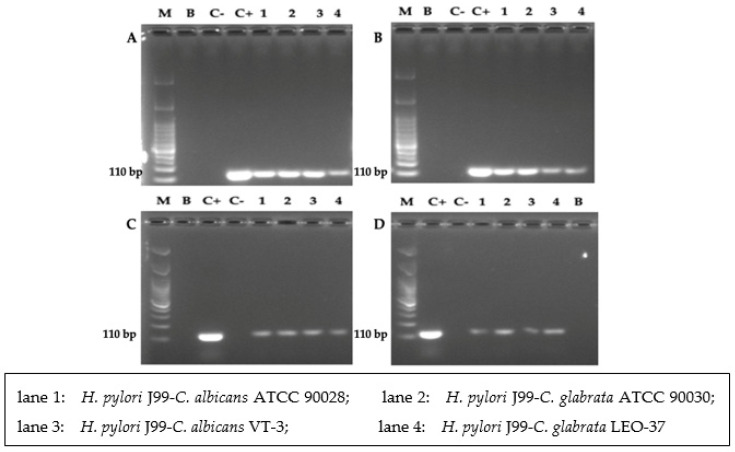
Image of a 2% agarose gel showing the amplicons obtained after amplifying the *H. pylori 16S rRNA* gene from total DNA extracted from *Candida* cells co-cultured with the different strains of *H. pylori* cultured in BB supplemented with 1%, 5% or 20% FBS or in SS. M: molecular weight markers, B: blank (master mix, primers, PCR grade water), C−: negative control (DNA from pure *C. glabrata* ATCC 90030), C+: positive control (DNA from pure *H. pylori* J99). Lanes 1–4 correspond to the amplicons obtained from DNA extracted from yeasts co-cultured with *H. pylori*. (**A**) Co-cultures in BB-1%FBS; (**B**) co-cultures in BB-5%FBS; (**C**) co-cultures in BB-20%FBS; (**D**) co-cultures in SS.

**Figure 11 biology-10-00426-f011:**
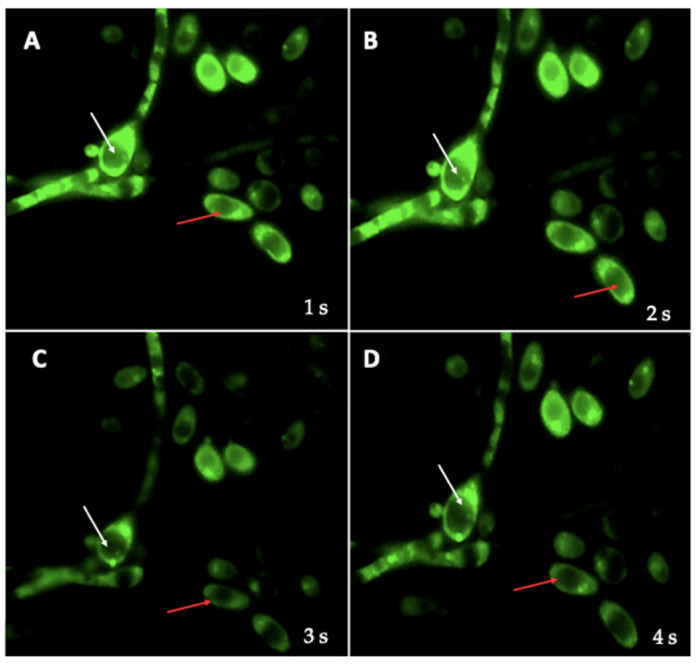
A, B, C and D correspond to a same microscopic field photographed at 1 s intervals. Cell viability, indicated by green fluorescence, of *H. pylori* harbored within *C. albicans* VT-3 after co-culturing them during 48 h. White arrows indicate yeast cells harboring intravacuolar *H. pylori*. The change in the position of *H. pylori*, at 1 s intervals, within the yeast can be observed (white arrow). In addition, the figure also shows yeast cells lacking intracellular *H. pylori* (red arrows).

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
