# Peer review of "Nutrient Deficiency Promotes the Entry of Helicobacter pylori Cells into Candida Yeast Cells"

_biology, 2021, doi:10.3390/biology10050426_

Round 1

Reviewer 1 Report

This manuscript by Kimberly Sanchez-Alonzo et al. describe data on the impact of nutrients deficiency on the entry of Hp into Candida cells..

This manuscript suffers from minor comments, I would recommend publication thereafter.

Methods : 

Whhy have the authors decided to include a yeast strain sampled in a vagina? The adaptation is clearly different so could have an impact on the results.

In general, how were selected the clinical strains for candida yeasts and Hp strains?

How was determine the MOI for bacteri-fungi co-culture?

For BLSB research, how many colonies have been selected? Why?

Give the version for ImageJ software.

"et al." must be italicized.

Author Response

Dear reviewer, I am attaching the answer to your comments.

Reviewer 2 Report

Thank you for submitting the manuscript “Nutrients deficiency promotes the entry of Helicobacter pylori cells into Candida yeast cells” to Molecules. The hypothesis raised in the work is interesting and current and deserves to be considered for publication. The introduction justifies why it is necessary to study the environment that promotes the survival of Helicobacter pylori cells. The methods used are consistent with the objective of the work.

The text in general has a language problem and needs an extensive revision of English.

For example:

L24 “the absence of nutrients intra-yeast bacteria were reduced” was

L32 “nutrients concentration” nutrient concentration

L37 “were present” was

Author Response

(The authors gave the same response as above.)

Reviewer 3 Report

Summary

In this study titled “Nutrients deficiency promotes the entry of Helicobacter pylori cells into Candida yeast cells”, the authors investigate in vitro if variations in nutrients concentration in the cultured medium trigger internalization of H. pylori within Candida cells.

They used four candida strains and four H. pylori strains and show that reduced nutrients stresses H. pylori promoting its entry into well fed Candida cells. Starvation of both H. pylori and Candida strains reduced H. pylori entry. In addition, H. pylori entry into Candida is dependent on the bacterial strains.

In principle an interesting study, specially considering the impact of nutrition. However, several points appeared preventing more positive review:

Major points:

Comment 1

Aren't these sentences a contradiction in terms? Line 23/24/25 “In fact, deficiency of nutrients increased the harboring of viable H. pylori cells within yeast cells. In the absence of nutrients intra-yeast bacteria were reduced.” Reading the entire paper I realized what the authors meant but they need to re-phrase.

Comment 2

The authors introduced H. pylori quite well but missed same care for introduction of Candida. In fact, where do both meet (H. pylori is mainly an anaerobic pathogen of the stomach while Candida colonizes the intestines, skin and mucous membranes). While H. pylori feels comfortable in acidic pH, candida mostly occurs in neutral pH. How does this fit?

So please adjust introduction.    

Comment 3

To me the discussion is not sufficient. In parts it repeats or reflexes the result section (e.g. line 511 ff). The authors should take more care in discussing the impact of their findings with respect to actual literature including e.g. “Adhesion of Helicobacter pylori to yeast cells by Ansorg R, Schmid EN. Zentralbl Bakteriol. 1998 Dec;288(4):501-8.” stating that “there was no indication that a direct cooperation with yeasts plays a role in H. pylori infections.”

Comment 4

I do not understand why 1% FBS led to high level of H. pylori entering Candida while 0% did not. I would have expected that there would be an effect as well. If iron was missing as discussed it should be easy to show data using SS supplemented with iron. Also 0.1% and 0.5% FBS should have been tested to get a better insight. Definitively this part should be considered and at least discussed more carefully.  

Comment 5

The cell viability tests should be expanded by testing if H. pylori coming from the inside of Candida are still able to proliferate. Such assay would definitively increase the impact of this manuscript and teach us the real relevance of the internalization of H. pylori into Candida. Even a kinetic answering how long H. pylori can survive in candida would be great.

Comment 6

The authors utilized four well established H. pylori strains and a fairly new isolate. Unfortunately they did not comment on the potential difference of strains culture since long in vitro culture ones should be more adapted than new ones considering that each H. pylori strains is unique.  

Comment 7

Line 501: “grew in all FBS concentrations tested” but Figure 4 shows also some growing even without FBS (=SS). This was not mentioned/discussed.

Comment 8

The authors need a figure summarizing the effects of four H. plyori strains and four Candida strains.

Comment 9

The following questions should be addressed:

Does H. pylori proliferate within candida (anaerobic condition, sufficient nutrition)?

Does Candida proliferate having incorporated H. pylori?

Does Candida proliferate having released incorporated H. pylori?

Comment 10

Figure 3 D should show the same scale on the y axes as Figure 3A, B, C to make it comparable at the first view.

Comment 11

Supplementary Figure S1, S2 and S3 should show the same scale on the y axes to make it comparable at the first view.

Comment 12

The authors should state the speed of centrifugation by x g and definitively not as 10,000 rpm (line 162, 193, 195).

Comment 13

Figure 1 which strain? Representative for the other strains? All should be shown at least in supplementary figures. Same counts for Figure 2 showing one representative? Candida strain.

Comment 14

Please consider better colors for Figure 6 and 7 because latest printed in black/white you can not any longer distinguish.

Comment 15

There are far too many repetitions/ abbreviations in the figure legends that should be reproduced elsewhere (Figure 10 starting from line 444, Figure 3 starting from line 292, etc.

Author Response

(The authors gave the same response as above.)

Round 2

Reviewer 3 Report

Fine with me. The authors answered more or less all issues I did raise and I do understand that not all is possible to do during this pandemic times. 

Author Response

Dear Reviewer :

Thank you very much for all your comments. Your comments allowed us to improve the manuscript.

Sincerely

The authors